# Winter Wheat Mapping Based on Sentinel-2 Data in Heterogeneous Planting Conditions

**Dongyan Zhang [1], Shengmei Fang [1], Bao She [1,2,*], Huihui Zhang [1,3] , Ning Jin [1,4], Haoming Xia [5] , Yuying Yang [1] and Yang Ding [1]**

[1] National Engineering Research Center for Agro-Ecological Big Data Analysis & Application, Anhui University, Hefei 230601, China; zhangdy@ahu.edu.cn (D.Z.); P18301126@stu.ahu.edu.cn (S.F.); Huihui.Zhang@usda.gov (H.Z.); jinn.13b@igsnrr.ac.cn (N.J.); P17201110@stu.ahu.edu.cn (Y.Y.); dingyangjd@stu.ahu.edu.cn (Y.D.)

[2] School of Geodesy and Geomatics, Anhui University of Science and Technology, Huainan 232001, China

[3] Water Management and Systems Research Unit, USDA Agricultural Research Service, Fort Collins, CO 80526, USA

[4] Department of Resources and Environment, Shanxi Institute of Energy, Jinzhong 030600, China

[5] Key Laboratory of Geospatial Technology for Middle and Lower Yellow River Regions (Henan University), Ministry of Education, Kaifeng 475004, China; xiahm@vip.henu.edu.cn

\* Correspondence: 2017044@aust.edu.cn; Tel.: +86-13858044819

**Abstract:** Monitoring and mapping the spatial distribution of winter wheat accurately is important for crop management, damage assessment and yield prediction. In this study, northern and central Anhui province were selected as study areas, and Sentinel-2 imagery was employed to map winter wheat distribution and the results were verified with Planet imagery in the 2017–2018 growing season. The Sentinel-2 imagery at the heading stage was identified as the optimum period for winter wheat area extraction after analyzing the images from different growth stages using the Jeffries–Matusita distance method. Therefore, ten spectral bands, seven vegetation indices (VI), water index and building index generated from the image at the heading stage were used to classify winter wheat areas by a random forest (RF) algorithm. The result showed that the accuracy was from 93% to 97%, with a Kappa above 0.82 and a percentage error lower than 5% in northern Anhui, and an accuracy of about 80% with Kappa ranging from 0.70 to 0.78 and a percentage error of about 20% in central Anhui. Northern Anhui has a large planting scale of winter wheat and flat terrain while central Anhui grows relatively small winter wheat areas and a high degree of surface fragmentation, which makes the extraction effect in central Anhui inferior to that in northern Anhui. Further, an optimum subset data was obtained from VIs, water index, building index and spectral bands using an RF algorithm. The result of using the optimum subset data showed a high accuracy of classification with a great advantage in data volume and processing time. This study provides a perspective for winter wheat mapping under various climatic and complicated land surface conditions and is of great significance for crop monitoring and agricultural decision-making.

**Keywords:** random forest; feature selection; planet; surface fragmentation

## 1. Introduction

Wheat (Triticum Aestivum L.) is the third-largest food crop in terms of production globally [1], providing a large number of nutritional sources for those suffering from nutrient deficiency. China is one of the major wheat-producing areas. In 2017, the wheat planting area in China reached 24,510 *kha* ranking third in the world, and more than 98% of the total acreage is winter wheat [2]. Therefore, it is

of great significance for the government to obtain accurate information on the planting area, growth and yield of winter wheat for formulating agricultural policies, estimating crop yield and ensuring food security [3]. The traditional method to acquire the planting area of winter wheat is mainly a manual sampling survey [4], which is not only labor-intensive, time-consuming and expensive but also susceptible to subjective factors. Remote-sensing technology has been widely used in the field of crop identification and mapping [5–7] due to its spatial coverage, temporal resolution, availability at near real-time and low cost.

The development of various satellites makes it possible to monitor cropping areas at fine spectral, temporal and spatial scales [8]. Moderate-resolution imaging spectroradiometer (MODIS) can provide a large amount of observation data that are both valuable and obligatory for global vegetation monitoring. It is sufficient to map large scale cropping areas [9,10], however, it is challenged by mixed pixels when the medium-spatial-resolution satellite data were used for crop characterization in finer-scale land distribution, such as southern China [11]. Recently, high-spatial-resolution satellites such as QuickBird and SPOT5 have become available and provide new opportunities for more accurate mapping of crops [12,13]. High-resolution data can provide more details of land tenure system, and improve accuracy of mapping [14]. However, the cost of high-spatial-resolution data limits their general use. With the successful launch of the second Sentinel-2 satellite in March 2017, Sentinel-2 mission proposed by European Space Agency is generating unprecedented volumes of data at high spatial (up to 10 m), spectral (13 bands) and temporal resolutions (minimum five day), which makes it used widely in crop identification and mapping [15–17].

Using remote-sensing data to monitor crop is mainly based on spectral and crop phenological difference, so the identification methods can be divided into spectral methods, such as remote-sensing-based classification [13,18], mixed pixel decomposition [19], multi-source information integration [7] and phenological method, such as time series analysis [8,20–22]. There are some other methods like remote-sensing sampling [23], random forest and support vector machine [24]. With the development of remote-sensing technology, more algorithms have been applied to crop identification. However, such study on winter wheat extraction is still limited in the areas which have difficulties and challenges such as [11]: unfavorable weather conditions; frequent cloud cover; complex terrain surface; high degree of fragmentation of cultivated land; and significant difference in crop planting structure between regions. Existing studies on winter wheat extraction in the areas with such conditions still have some shortcomings: 1) the methods were simple and the accuracy was not high; 2) few have focused on the remote-sensing mapping of wheat in the areas due to the high degree of surface fragmentation; and 3) few have discussed the differences of extraction methods under different planting conditions.

In 2017, the planting area of winter wheat in Anhui province reached 2822.2 *kha*, ranking the third in China after Henan and Shandong provinces [2]. However, the application of remote-sensing technology in Anhui agricultural production is relatively insufficient compared with other provinces due to the difficulties mentioned above. In view of the important role of Anhui province in China's wheat production and the problems in the remote-sensing extraction of winter wheat, it is urgent to explore better methods for winter wheat mapping using remote-sensing imagery.

Therefore, the objective of the study was to explore the effect of different planting conditions (climate, scale, fragmentation) on the accuracy of winter wheat mapping using remote-sensing data. The specific goals were to (1) select the optimum phenological phase for extracting winter wheat from Sentinel-2 images based on key phenological periods; (2) acquire the optimum remote-sensing screening features of winter wheat the data acquired in the optimum phenological period; (3) get the optimum scheme for winter wheat mapping in the areas of interest; and (4) identify the uncertainties and future needs in winter wheat mapping.

## 2. Materials

### 2.1. Study Area

Anhui province abuts the Yangtze River Delta Economic Zone centered in Shanghai. The province is located in the middle and lower reaches of the Yangtze and Huaihe Rivers and belongs to the East China region (114°54′~119°37′E, 29°41′~34°38′N) (Figure 1). Affected by monsoon, Anhui has four distinct seasons and is a transitional region between warm temperate zone and subtropical zone [25]. The topography and landforms are diverse, with plains, hills and low mountains [26].

Northern and central zones are the main producing areas of winter wheat in Anhui [27]. Typical counties for winter wheat production in the two areas were selected as study areas due to the differences in climatic conditions, topography and surface fragmentation.

Lingbi county and Sixian county were selected in northern Anhui (Northern Anhui Counties, NAC). The cultivated land area in northern Anhui accounts for about half in the whole province. It is mainly flat plain with mild and semi-humid climate, frequent drought and flood disasters, and affected by climate change. The annual mean precipitation and annual mean temperature were about 843 mm and 15.4 °C in the area, respectively [27]. The winter wheat planting area in northern Anhui accounts for about 70% of the province's total amount [28].

Changfeng county and Dingyuan county were chosen in central Anhui (Central Anhui Counties, CAC). Central Anhui belongs to a hilly landform with a mild climate, moderate rainfall and sufficient sunshine. During the study period, the annual mean precipitation was about 951 mm, and the annual mean temperature was 16.2 °C [27]. The difference in climate between northern and central zones results in huge variations in crop types. The challenge of wheat extraction in this region is greater than the northern area because of the complex planting structures and discontinuous wheat fields. Generally, winter wheat is planted in late October and harvested in early June in the following year and the winter wheat phenology in NAC and CAC (http://data.cma.cn/site/index.html) is shown in Table 1.

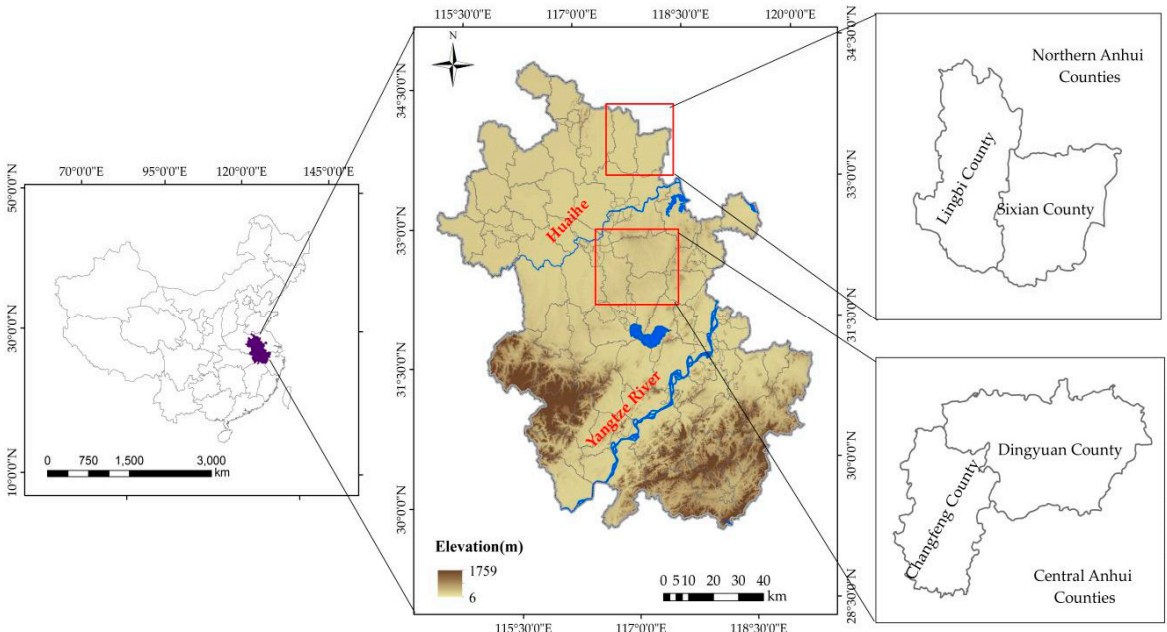

**Figure 1.** Maps of the study areas.

**Table 1.** Phenological stages of winter wheat in northern Anhui counties (NAC) and central Anhui counties (CAC) (2017–2018).

|  | NAC | CAC |
|---|---|---|
| Sowing | 24 October, 2017 | 28 October, 2017 |
| Seeding | 6 November, 2017 | 9 November, 2017 |
| Tillering | 13 December, 2017 | 14 December, 2017 |
| Overwintering | 25 December, 2017 | 2 January, 2018 |
| Greening up | 14 February, 2018 | 17 February, 2018 |
| Jointing | 21 March, 2018 | 12 March, 2018 |
| Heading | 14 April, 2018 | 12 April, 2018 |
| Maturing | 15 May, 2018 | 10 May, 2018 |
| Harvesting | 23 May, 2018 | 27 May, 2018 |

*2.2. Datasets*

Sentinel-2 satellite carries a multi-spectral imager (MSI) with an orbital altitude of 786 km, 13 spectral bands, a swath width of 290 km and four bands at 10 m, six bands at 20 m and three bands at 60 m spatial resolution. There are three bands in the red-edge spectral region providing more band choices for vegetation monitoring [16,29,30]. Planet has a spatial resolution of 3 meters and four reflective bands (Blue, 0.45–0.51 μm; Green, 0.50–0.59 μm; Red, 0.59–0.67 μm; NIR, 0.78–0.86 μm). It can update global data once a day (after 2016) [31] and provide guarantee for change monitoring of different frequencies, and it has an advantage in price compared with other satellite data with similar resolution worldwide [32].

In this study, the images were selected by referring to the average phenological period of wheat (http://data.cma.cn/site/index.html) in the area of interest and considering the coverage of available images. We employed five Sentinel-2 images in each of the two study areas and purchased eleven Planet images (each for 24×7 km$^2$) (http://www.kosmos-imagemall.com/) (Table 2).

**Table 2.** Description of the satellite data used in the study.

| Sensor | Acquisition Time | The Number of Scenes | Wheat Growth Stage |
|---|---|---|---|
| Sentinel-2 MSI | 8 November 2017 | 2 | Seeding |
| Sentinel-2 MSI | 18 December 2017 | 2 | Tillering |
| Sentinel-2 MSI | 11 February 2018 | 2 | Greening up |
| Sentinel-2 MSI | 7 April 2018 | 2 | Heading |
| Sentinel-2 MSI | 6 June 2018 | 2 | Harvesting |
| Planet CCD | 7 April 2018 | 11 | Heading |

## 3. Methods

*3.1. Data Preprocessing*

Sentinel-2 images in the winter wheat growing season of 2017–2018 were downloaded from the European Space Agency (ESA) (https://scihub.copernicus.eu/). For the remainder of analysis, we only focused on Sentinel-2 10 m and 20 m bands. Using ESA Sen2Cor plugin v2.5.5 processor (http://step.esa.int/main/third-party-plugins-2/sen2cor/), available on the Sentinel Application Platform (SNAP) (http://step.esa.int/main/download/), we processed reflectance images from Top-of-Atmosphere (TOA) Level 1C Sentinel-2 to Bottom-of-Atmosphere (BOA) Level 2A. A nearest neighbor-based re-sampling to 10 m spatial resolution was performed to bands 5, 6, 7 and 8a to achieve the same resolution of bands 2, 3, 4 and 8. Finally, the ten bands were imported in ENVI (https://www.harrisgeospatial.com/), stacked and cropped over the area of interest.

### 3.2. Calculating the Separability Using Jeffries–Matusita (JM) Distance

Previous studies have shown that Jeffries–Matusita (JM) distance is an effective metric to evaluate the separability of training samples in remote-sensing-based classification [8,33,34]. We calculated the separability between winter wheat and other land cover types by JM distance to determine the optimum period imagery.

In NAC, we selected seven main land cover types (winter wheat, water, urban, bare land, grass, forest and others) with 50 training sets for each type based on five key phenological period images. Oilseed rape and barley only accounted for 1.5% of the area planted for winter crops, which was ignored [27]. In CAC, there was little barley in winter (3.0%), but more oilseed rape (26%) [27]. we chose eight land cover types (winter wheat, water, urban, bare land, grass, forest, oilseed rape and others) with about seventy training sets for each type.

The *JM* distance is calculated as:

$$JM(\omega_i, \omega_j) = \int_\chi \left[ \sqrt{p(\chi|\omega_i)} - \sqrt{p(\chi|\omega_j)} \right]^2 d\chi \tag{1}$$

where $x$ represents a random variable, $w_i$ and $w_j$ are the two land cover types under consideration. Under normality assumptions, Equation (1) can simplify to:

$$JM = 2(1 - e^{-\partial}) \tag{2}$$

$$\partial = \frac{1}{8}(\rho_i - \rho_j)^T \left(\frac{\sum i + \sum j}{2}\right)^{-1} (\rho_i - \rho_j) + \frac{1}{2} \ln\left[\frac{\left|\frac{(|\sum i + \sum j)|}{2}\right|}{(|\sum i\| \sum j|)^{\frac{1}{2}}}\right] \tag{3}$$

where, $\rho_i$ and $\rho_j$ are the averages of spectral reflectance of type-specific and $\sum i$ and $\sum j$ are estimates for the type-specific covariance matrices.

*JM* distance ranges between 0 (low separability) and 2 (high separability). Values of *JM* > 1.8 indicates good separability between two samples [35].

### 3.3. Description for Spectral Features

Nineteen features were selected in total (Table 3). The reflectance of ten bands were selected as spectral features based on the optimum period imagery of winter wheat while reflectance values were used to calculate the normalized difference vegetation index (NDVI) [36], enhanced vegetation index (EVI) [37], soil-adjusted vegetation index (SAVI) [38], greenness normalized difference vegetation index (GNDVI) [39], modified normalized difference water index (MNDWI) [40] and normalized difference building index (NDBI) [41]. Previous research stressed the importance of the red-edge bands [16,42,43], so three red-edge indices ($NDVI_5$, $NDVI_6$ and $NDVI_7$) [44] were calculated from the three red-edge bands.

**Table 3.** Information about features [1].

| Features | Band Number | Instructions | | Reference |
|---|---|---|---|---|
| | | Center wavelength (nm) | Resolution (m) | |
| | band 2 | 496 | 10 | |
| | band 3 | 560 | 10 | |
| | band 4 | 665 | 10 | |
| | band 5 | 704 | 20 | |
| **Spectra** | band 6 | 740 | 20 | - |
| | band 7 | 783 | 20 | |
| | band 8 | 835 | 10 | |
| | band 8a | 865 | 20 | |
| | band 11 | 1614 | 20 | |
| | band 12 | 2202 | 20 | |

**Table 3.** *Cont.*

| Features | Band Number | Instructions | Reference |
|---|---|---|---|
| Vegetation indices | NDVI | $(\varrho_8 - \varrho_4)/(\varrho_8 + \varrho_4)$ | Rouse et al., 1974 |
| | NDVI$_5$ | $(\varrho_8 - \varrho_5)/(\varrho_8 + \varrho_5)$ | Gitelson, 1997 |
| | NDVI$_6$ | $(\varrho_8 - \varrho_6)/(\varrho_8 + \varrho_6)$ | Gitelson, 1997 |
| | NDVI$_7$ | $(\varrho_8 - \varrho_7)/(\varrho_8 + \varrho_7)$ | Gitelson, 1997 |
| | EVI | $2.5 \times ((\rho_8 - \varrho_4)/(\varrho_8 + 6\varrho_4 - 7.5\varrho_2 + 1))$ | Huete et al., 2002 |
| | SAVI | $1.5 \times ((\varrho_8 - \varrho_4)/(\varrho_8 + \varrho_4 + 1))$ | Huete et al., 1988 |
| | GNDVI | $(\varrho_8 - \varrho_3)/(\varrho_8 + \varrho_3)$ | Gitelson, 1997 |
| Water index | MNDWI | $(\varrho_3 - \varrho_{11})/(\varrho_3 + \varrho_{11})$ | Xu, 2005 |
| Building index | NDBI | $(\varrho_{11} - \varrho_8)/(\varrho_{11} + \varrho_8)$ | Zha, 2003 |

[1] $\varrho_2$, $\varrho_3$, $\varrho_4$, $\varrho_5$, $\varrho_6$, $\varrho_7$, $\varrho_8$ and $\varrho_{11}$ represent the spectral reflectance of bands 2, 3, 4, 5, 6, 7, 8 and 11, respectively.

### 3.4. Description of the Classification Scheme

Four schemes of classification (Table 4) were designed for two purposes [45]: 1) examine the influence of different features on winter wheat extraction and determine their importance; and 2) explore the optimum schemes of winter wheat extraction in the area of interest. Scheme D was generated by the result of feature selection.

**Table 4.** Design of experimental scheme.

| Scheme | Spectral Feature Combination |
|---|---|
| A | Nine indices (seven vegetation indices + water index+ building index) |
| B | Ten spectral bands (bands 2, 3, 4, 5, 6, 7, 8, 8a, 11, 12) |
| C | Nineteen features (ten spectral bands + nine indices) |
| D | Optimum subset from the nineteen features |

### 3.5. Random Forest Algorithm for Selecting Features and Extracting Winter Wheat

Random forest (RF) is a popular algorithm for classification and feature selection [46,47]. Recently, random forest has been widely used in many fields because of its high classification accuracy, strong anti-noise and anti-outlier ability. Moreover, the variable importance metric can be used as an effective tool for feature selection [48,49].

Not all features from imagery are useful in improving classification accuracy. It is a key step in how to choose the most important features for crop identification in image analysis process [4]. Random forest not only can realize remote-sensing-based classification but also plays an important role in feature selection [50,51]. The sample-set was selected based on optimum period imagery for wheat extraction in this study. Samples about three hundred evenly composed of winter wheat and other mainland cover types (water, urban, bare land, grass, forest, oilseed rape and others) were selected as the original classification samples in two study areas through visual interpretation with the help of Google Earth and Planet images. Two-thirds of the training samples were randomly selected from the original sample set. The attribute of the training samples and the features participating classifications were used to train the classifier. The remaining one-third of the unsampled samples were called out-of-bag (OOB) data [47,50]. Out-of-bag-error generated by OOB data can evaluate the classification ability of the classifier and calculate the variables' importance (VI) for feature selection. The variables importance score of feature $j$ is calculated as [51]:

$$VI_j = \frac{1}{N}\sum_{i=1}^{N} (A^j_{Ni} - A^j_{Oi}) \tag{4}$$

where $VI_j$ represents the importance score of feature $j$, $N$ represents the number of decision trees generated, $A^j_{Ni}$ represents the OOB error of decision tree $i$ when noise is not added to feature $j$ and $A^j_{Oi}$

represents the OOB error of decision tree *i* when noise is added to feature j. If the accuracy of OOB is reduced greatly, it indicates that feature *j* has a great influence on the classification results after adding noise to it, that is to say, its importance is high.

In order to determine the influence of different feature variables (Table 3) on the extraction of winter wheat, we applied a random forest algorithm to score them and assess the importance of different features, which was realized by the random forest package in MATLAB 2018 (https://ww2.mathworks.cn/products/matlab.html). There are two important parameters in the random forest function: the number of the decision tree: ntree, and the number of features selected by each split node: mtry. In this study, mtry = $\sqrt{N}$ is the default value [47,50] (N represents the number of all features). Theoretically, the larger the ntree is, the higher the accuracy of classification will be, but the computation and time cost will increase. We found that when ntree was greater than 100, the OOB error tends to be stable under the default condition of mtry, so the ntree parameter was set as 100 [52].

### 3.6. Accuracy Assessment

The results were tested from two perspectives of area extraction accuracy and spatial distribution. Percentage error (PE) was employed to quantify the difference of wheat mapping areas between results from Planet and results from Sentinel-2 as:

$$PE = \frac{|Reference - Estimated|}{Reference} \times 100\% \qquad (5)$$

where *Reference* represents the wheat area extracted from each Planet sample plot. *Estimated* represents the area extracted from each Sentinel-2 sample plot.

Confusion matrix [53] is a standard means to evaluate the accuracy of classification results from remote-sensing images, including the producer's accuracy (PA), user's accuracy (UA) and Kappa coefficient (Kappa). *Kappa* was calculated as [8,54]:

$$Kappa = \frac{N\sum_{i=1}^{r} x_{ii} - \sum_{i=1}^{r}(x_{i+} \times x_{+i})}{N^2 - \sum_{i=1}^{r}(x_{i+} \times x_{+i})} \qquad (6)$$

where *r* is the number of rows, $x_{ii}$ is the number of pixels in row *i* and column *i*, $x_{i+}$ is the total number of pixels in row *i*, $x_{+i}$ is the total number of pixels in column *i*, and *N* is the total number of pixels.

Five sample plots (each for $6 \times 6$ km$^2$) in NAC (Figure 2a) and six sample plots (each for $6 \times 6$ km$^2$) in CAC (Figure 2b) based on Planet imagery were selected for evaluating the result derived from Sentinel-2. Since winter wheat at the heading stage is conducive to visual interpretation, this study applied Planet images (3 m) at the heading stage to extract distribution of winter wheat in sample plots, and the results were used as reference data to verify the accuracy. Specifically, Planet imagery was subset according to the sample plot size, then four spectral bands of Planet were employed as input to obtain the distribution of winter wheat by using the random forest algorithm. The accuracy of the winter wheat extraction in Sentinel-2 sample plots was validated by referring to the distribution of winter wheat in Planet sample plots. Since Planet's spatial resolution is 3 m and sentinel-2 is 10 m, all results from Sentinel-2 were re-sampled to 3 m, using the nearest algorithm to match the spatial resolution of Planet to achieve spatial corresponding for each pixel.

Five main steps were performed for winter wheat extraction and validation: 1) preprocessing data; 2) screening the optimum period based on five key phenological periods of winter wheat growing seasons; 3) screening the optimum feature for winter wheat extraction; 4) identifying the optimum scheme for winter wheat mapping; and 5) evaluating accuracy with Planet imagery. The workflow applied in this study is shown in Figure 3.

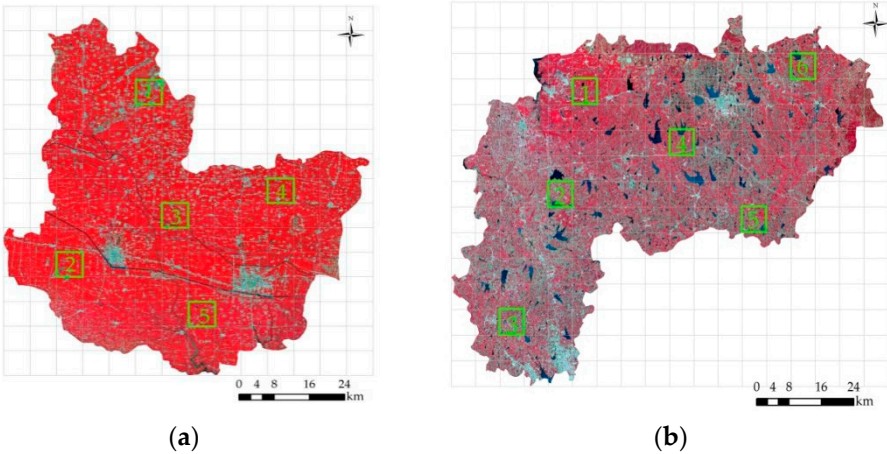

(**a**)                                              (**b**)

**Figure 2.** The distribution of sample plots derived from Planet images. (**a**) Northern Anhui counties (NAC). (**b**) Central Anhui counties (CAC). (The base images are Sentinel-2 image on 7 April 2018 illustrated in false-color composite (R: NIR, G: Red, B: Green) and the points represent the sample plots number).

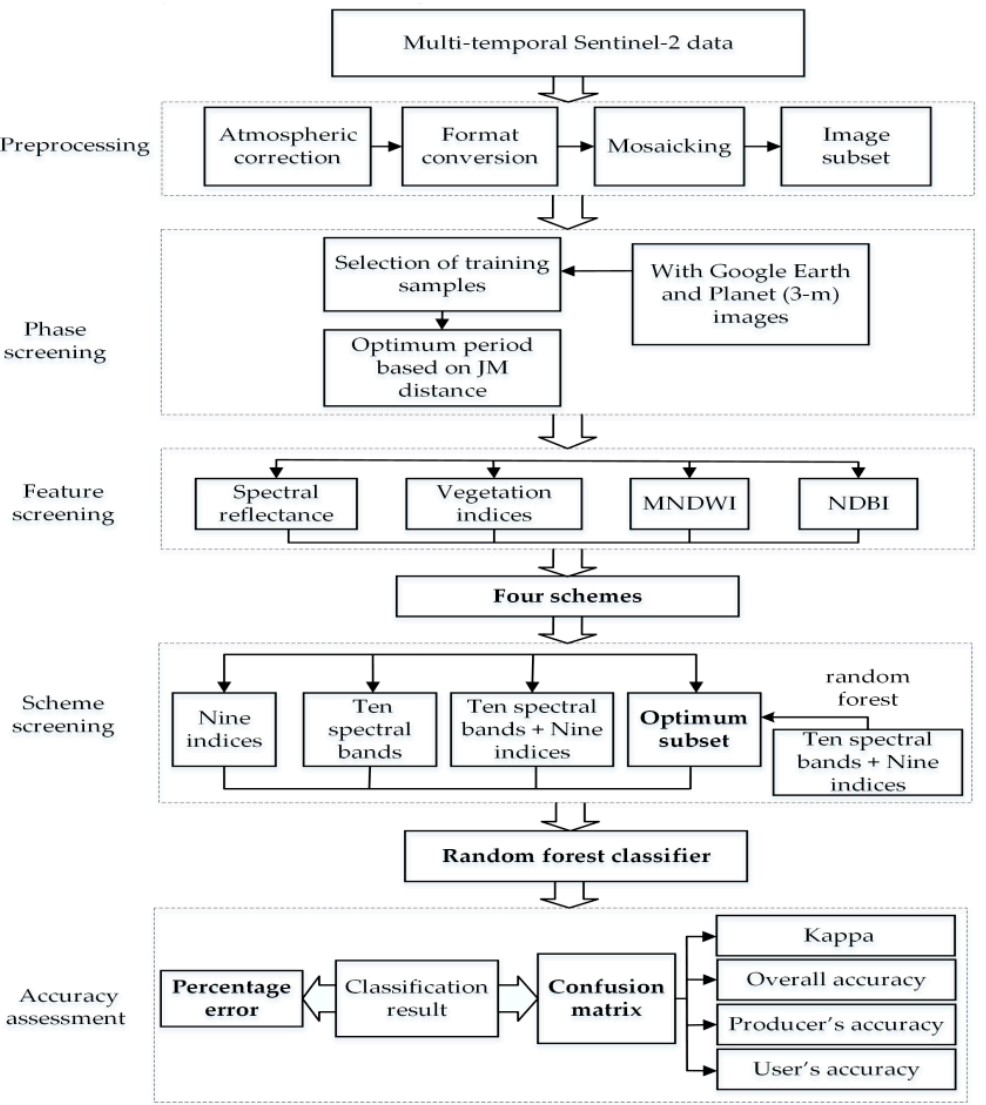

**Figure 3.** The workflow of the winter wheat extraction and validation.

## 4. Results

### 4.1. Selection of Optimum Periods

The separability was calculated using JM distance in NAC and CAC (Table 5). The result showed the separability between winter wheat and non-vegetation (water, urban, bare land, etc.) was good (JM > 1.8) in different phenological periods in both study areas. However, the separability between winter wheat and vegetation (forest, grass and oilseed rape) was slightly worse. From Table 5, we can see only on 7 April, 2018, the JM distance between winter wheat and other land cover types were all greater than 1.8, indicating that the heading stage was the optimum period to distinguish winter wheat from other land cover types.

**Table 5.** Jeffries–Matusita (JM) distance between winter wheat and other main land cover types.

| NAC | 8 November 2017 | 18 December 2017 | 11 February 2018 | 7 April 2018 | 6 June 2018 |
|---|---|---|---|---|---|
| Water | 1.98 | 1.99 | 1.99 | 1.99 | 1.99 |
| Urban | 1.99 | 1.99 | 1.99 | 1.99 | 1.99 |
| Bare land | 1.56 | 1.70 | 1.55 | 1.98 | 1.76 |
| Grass | 1.85 | 1.86 | 1.71 | 1.96 | 1.84 |
| Forest | 1.87 | 1.76 | 1.66 | 1.86 | 1.98 |
| Others | 1.83 | 1.80 | 1.62 | 1.99 | 1.92 |
| **CAC** | **8 November 2017** | **18 December 2017** | **11 February 2018** | **7 April 2018** | **6 June 2018** |
| Water | 1.97 | 1.99 | 1.97 | 1.99 | 1.98 |
| Urban | 1.97 | 1.99 | 1.97 | 1.99 | 1.97 |
| Bare land | 1.77 | 1.62 | 1.46 | 1.99 | 1.72 |
| Grass | 1.96 | 1.81 | 1.68 | 1.83 | 1.99 |
| Forest | 1.99 | 1.99 | 1.84 | 1.94 | 1.99 |
| Oilseed rape | 1.70 | 1.41 | 1.70 | 1.96 | 1.70 |
| Others | 1.99 | 1.99 | 1.97 | 1.99 | 1.83 |

### 4.2. Selection of Optimum Features

We scored the features (Table 3) generated from Sentinel-2 image acquired on 7 April 2018. The scores for each feature in NAC (Figure 4a) and in CAC (Figure 4b) were calculated by random forest algorithm.

The result showed: In NAC, the NDVI had the highest score (3.64), which was the key feature for winter wheat extraction, MNDWI had the lowest (0.07), so it had little impact. In CAC, band 6 scored the highest (5.57) while NDBI scored the lowest (0.26), so band 6 was the most important feature.

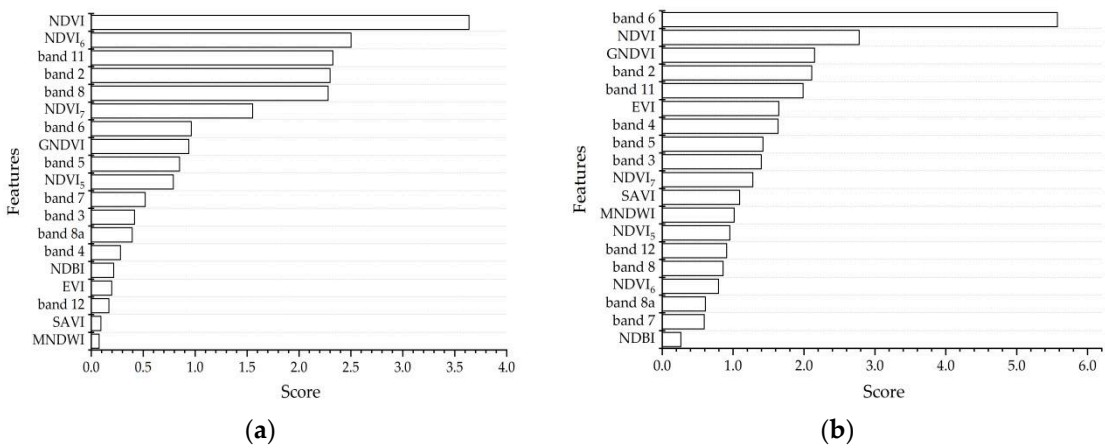

**Figure 4.** Score for importance of features generated from Sentinel-2 image on 7 April 2018. (**a**) Northern Anhui counties (NAC). (**b**) Central Anhui counties (CAC).

### 4.3. Winter Wheat Mapping in NAC and CAC

For the determination of scheme D, nineteen features were arranged in descending order according to the importance score (Figure 4). A feature with the lowest score was removed from the feature set by Sequential Backward Selection (SBS) [55], and a classification model was constructed with the remaining features. The number of optimum feature subsets was determined by the classification accuracy, which refers to the prediction accuracy of OOB data by the classifier (Figure 5).

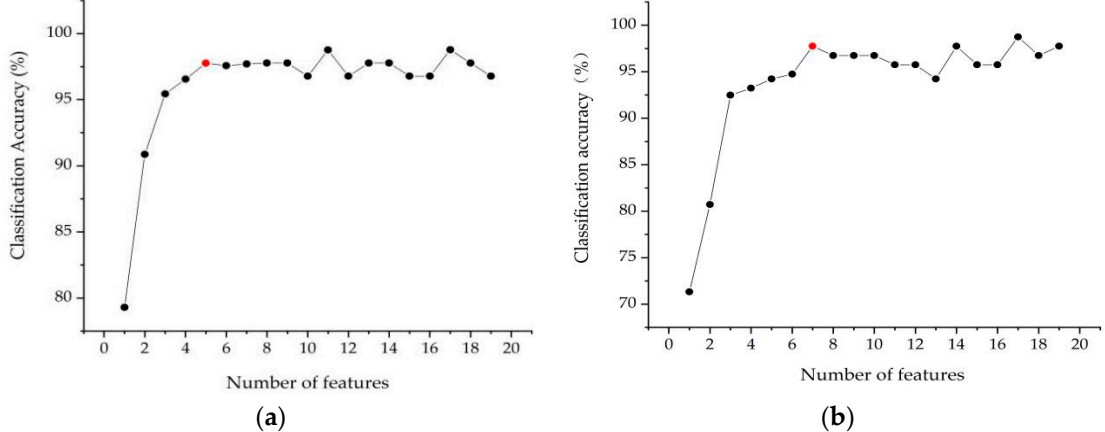

**Figure 5.** The relationship between the number of features and classification accuracy based on sequential backward selection. (**a**) Northern Anhui counties (NAC). (**b**) Central Anhui counties (CAC). (The red dot represents the number of optimum feature subset).

In NAC (Figure 5a), at first, less important features (features with the lowest score in importance score) were deleted, and the number of features decreased from 19 to 17. As a result, the classification accuracy increased generally, and the deletion of redundant features resulted in improved classifier performance. Secondly, the number of important features was reduced from 17 to 5. The classification accuracy changed little with the average classification accuracy of 96.86%, which indicated the classification accuracy remained stable with the reduction of the number of features participating in the classification. Finally, when the number of important features was reduced from 5 to 1, the classification accuracy was greatly reduced with the deletion of the features, which was caused by the elimination of useful features. In CAC (Figure 5b), there was a similar trend as in NAC. The classification accuracy fluctuated slightly with the reduction of the number of features from 19 to 7, and the classification accuracy decreased greatly when important features were deleted (7 to 1).

The classification accuracy with the first five features in NAC was 95.94% and that of the first seven features in CAC was 93.58% while the data volume was reduced by more than 60% compared with the original nineteen features. It effectively reduced the data volume and ensured a higher classification accuracy. Therefore, scheme D was done using the first five features (NDVI, NDVI$_6$, band 11, band 2 and band 8) presented in Figure 4a for NAC and the first seven features (band 6, NDVI, GNDVI, band 2, band 11, EVI and band 4) in Figure 4b for CAC.

Random forest classifier was used to extract the planting information of winter wheat in the study area of the four experimental schemes (Figure 6). In NAC (Figure 6a), the spatial distribution of winter wheat was continuous, and the scale of planting was large. In CAC (Figure 6b), the planting of winter wheat was relatively small and scattered, with fragmented landscape and small patches of farmland.

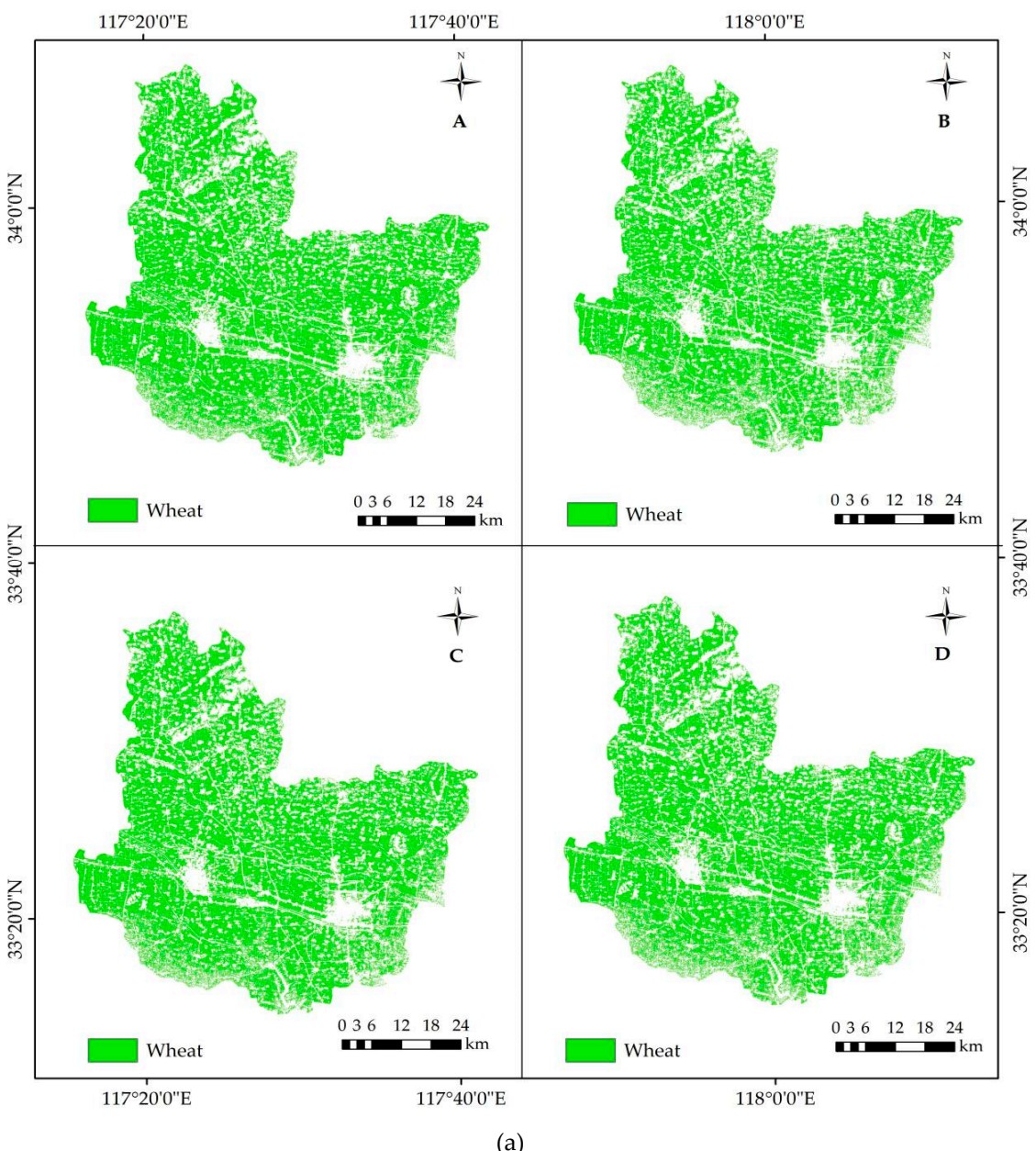

(a)

**Figure 6.** *Cont.*

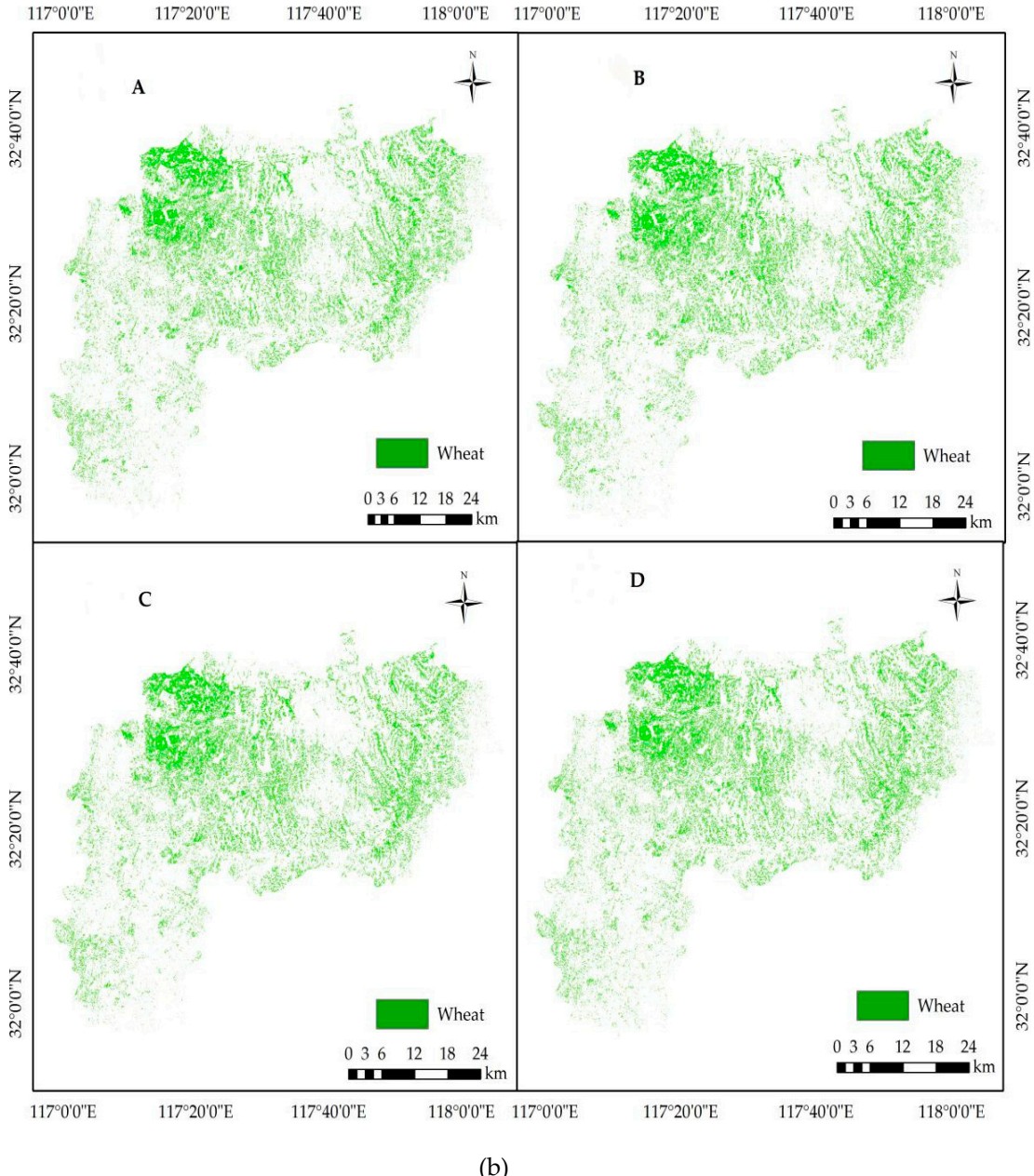

(b)

**Figure 6.** Four schemes for winter wheat mapping during the 2017–2018 growing season. (**a**) Northern Anhui counties (NAC). (**b**) Central Anhui counties (CAC). (A, B, C, D represent the wheat distribution based on scheme A, B, C, D).

### 4.4. Accuracy of Winter Wheat Maps

With Planet imagery extraction results as a reference, the five sample plots in NAC (Figure 2a) and six sample plots in CAC (Figure 2b) evenly distributed in the study area were employed to verify the accuracy of four schemes. The difference between the winter wheat distribution map by the Planet imagery and the map from Sentinel-2 by the four classification schemes for the sample plots in NAC and CAC are shown in Figure 7. It can be seen clearly that scheme A and B generated greater differences compared to scheme C and D, indicating that scheme C and D had better classification results.

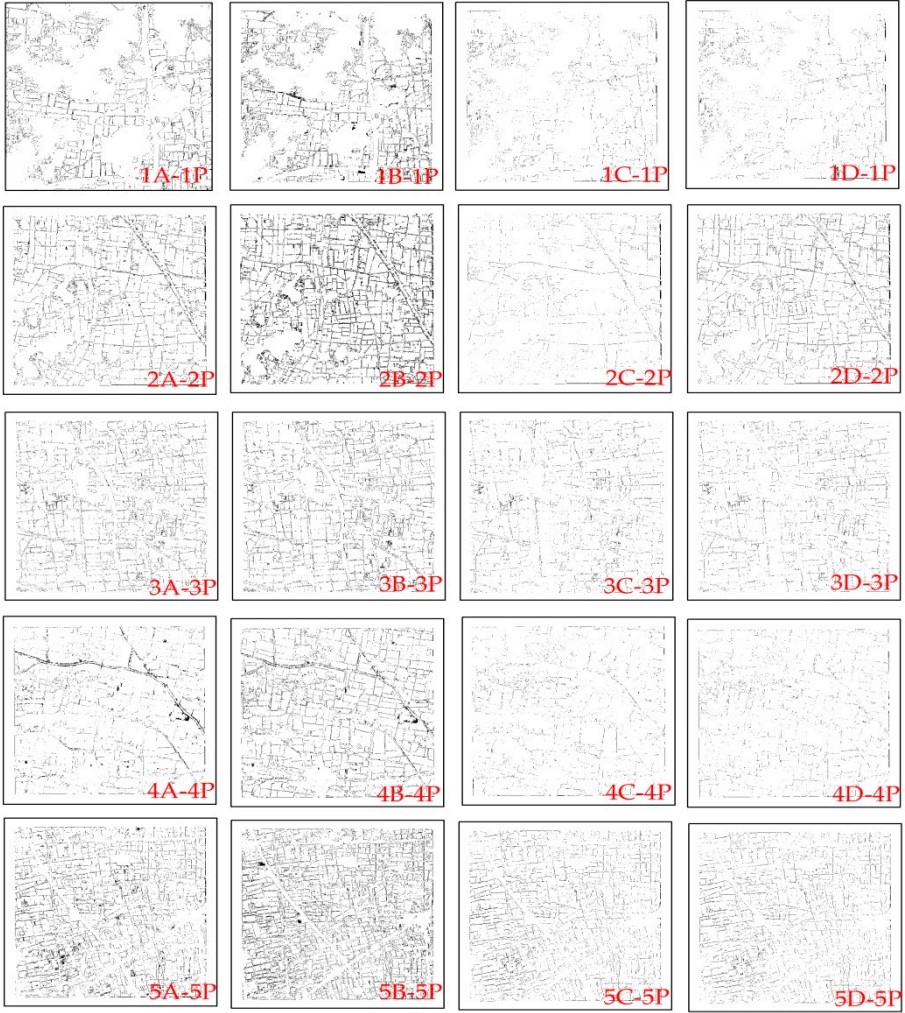

(a) NAC

**Figure 7.** *Cont.*

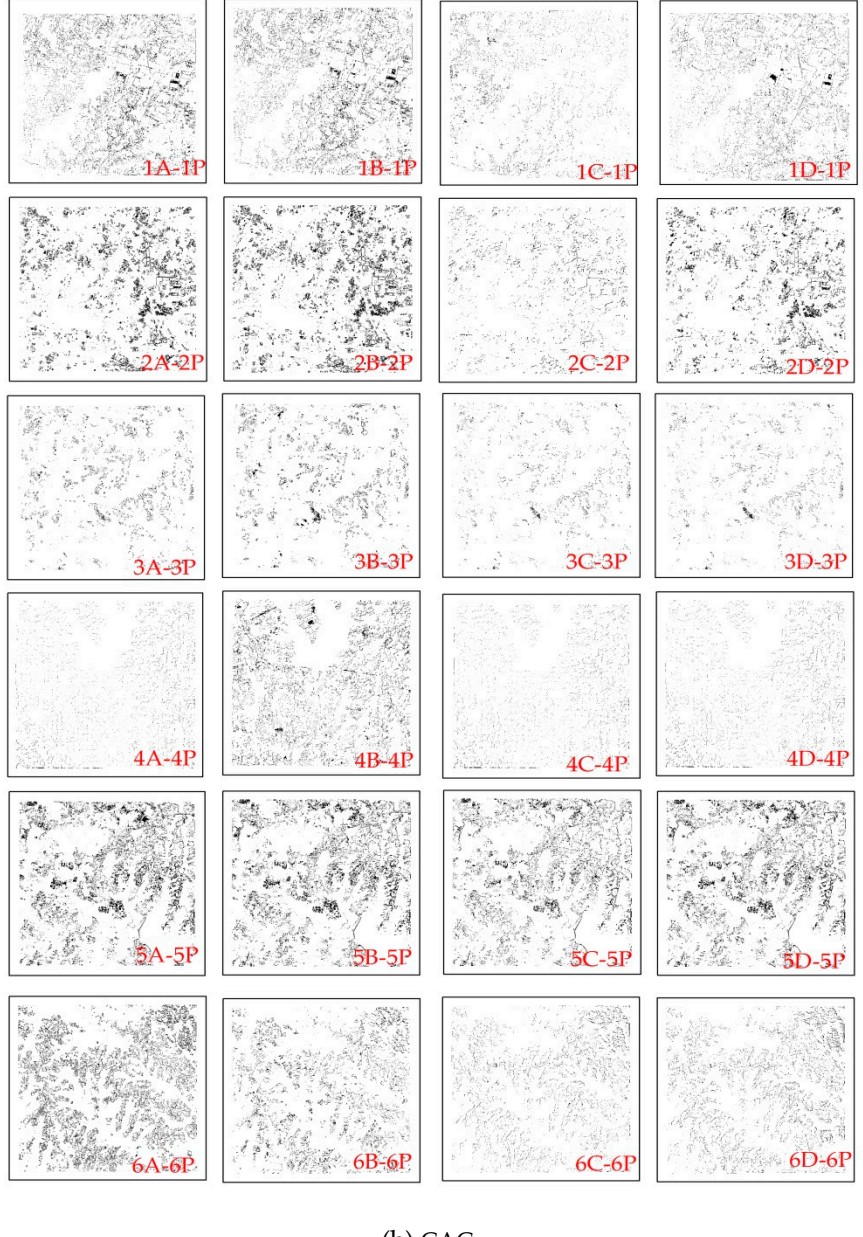

(b) CAC

**Figure 7.** Detailed distribution difference maps of winter wheat between Sentinel-2 and Planet imagery in the sample plots in NAC (**a**) and CAC (**b**). Black color indicates the difference between Sentinel-2 results and Planet results. 1–5 or 1–6 represents the sample plot number. P represents wheat distribution in the sample plots by the Planet imagery, A, B, C, D represent the wheat distribution in the sample plots based on scheme A, B, C, D.

From Table 6, the producer's accuracy and user's accuracy of the four wheat maps in NAC were relativity high ranging from 91% to 97% while that in CAC were from 72% to 92%. The Kappa coefficient was about 0.85 and percentage error (PE) was lower than 5% for NAC. The Kappa coefficient was between 0.7 and 0.8 and PE ranged from 12% to 30% in CAC. The overall extraction effect was inferior to that of the main producing areas in NAC. Moreover, it reflected in both study areas that the accuracy of scheme A and B was lower than that of scheme D, while the result of scheme C was slightly better (0.1 higher in Kappa) than D.

**Table 6.** Confusion matrix and percentage error (PE) for NAC and CAC [1].

| NAC | A | | | | B | | | | C | | | | D | | | |
|---|---|---|---|---|---|---|---|---|---|---|---|---|---|---|---|---|
| | PA (%) | UA (%) | Kappa | PE (%) | PA (%) | UA (%) | Kappa | PE (%) | PA (%) | UA (%) | Kappa | PE (%) | PA (%) | UA (%) | Kappa | PE (%) |
| Sample_1 | 94.7 | 91.4 | 0.85 | 3.6 | 90.3 | 91.7 | 0.85 | 3.2 | 94.5 | 91.5 | 0.88 | 3.0 | 94.1 | 91.0 | 0.87 | 3.2 |
| Sample_2 | 91.4 | 97.5 | 0.82 | 5.0 | 96.0 | 93.1 | 0.82 | 4.9 | 94.2 | 95.0 | 0.86 | 4.1 | 95.5 | 94.3 | 0.83 | 4.2 |
| Sample_3 | 95.6 | 96.7 | 0.84 | 2.0 | 94.6 | 96.3 | 0.85 | 2.2 | 95.5 | 96.9 | 0.88 | 1.4 | 95.4 | 96.4 | 0.87 | 1.5 |
| Sample_4 | 94.1 | 95.1 | 0.83 | 1.7 | 97.0 | 92.9 | 0.83 | 1.7 | 95.5 | 94.4 | 0.85 | 1.1 | 97.0 | 93.1 | 0.84 | 1.4 |
| Sample_5 | 94.6 | 96.3 | 0.85 | 3.4 | 91.2 | 97.7 | 0.82 | 5.6 | 94.1 | 91.0 | 0.87 | 1.3 | 95.1 | 96.7 | 0.86 | 2.7 |

| CAC | A | | | | B | | | | C | | | | D | | | |
|---|---|---|---|---|---|---|---|---|---|---|---|---|---|---|---|---|
| | PA (%) | UA (%) | Kappa | PE (%) | PA (%) | UA (%) | Kappa | PE (%) | PA (%) | UA (%) | Kappa | PE (%) | PA (%) | UA (%) | Kappa | PE (%) |
| Sample_1 | 76.0 | 91.6 | 0.73 | 16.9 | 84.1 | 85.7 | 0.77 | 15.4 | 79.4 | 90.3 | 0.78 | 12.0 | 76.0 | 91.6 | 0.76 | 16.4 |
| Sample_2 | 81.7 | 80.3 | 0.70 | 23.8 | 82.8 | 74.6 | 0.72 | 20.6 | 81.2 | 74.9 | 0.71 | 20.7 | 81.2 | 74.9 | 0.71 | 21.7 |
| Sample_3 | 81.9 | 66.9 | 0.72 | 21.6 | 75.1 | 65.3 | 0.69 | 27.6 | 81.8 | 66.1 | 0.72 | 20.8 | 82.2 | 68.3 | 0.73 | 18.6 |
| Sample_4 | 93.2 | 77.6 | 0.78 | 19.6 | 90.1 | 71.1 | 0.71 | 22.6 | 93.4 | 77.0 | 0.78 | 17.3 | 91.9 | 79.0 | 0.78 | 15.8 |
| Sample_5 | 72.6 | 65.7 | 0.70 | 23.1 | 84.4 | 73.2 | 0.71 | 22.4 | 76.7 | 84.3 | 0.72 | 20.8 | 74.9 | 72.1 | 0.70 | 24.4 |
| Sample_6 | 85.4 | 71.1 | 0.71 | 24.1 | 77.6 | 75.8 | 0.70 | 27.5 | 81.2 | 74.9 | 0.71 | 24.1 | 82.8 | 73.7 | 0.71 | 25.3 |

[1] PA: producer's accuracy; UA: user's accuracy; PE: percentage error. A, B, C and D represent scheme A, B, C and D.

## 5. Discussion

A range of winter wheat mapping approaches has been developed in previous studies based on remote-sensing imagery [4,7,8,10]. However, many of them have left problems in their research that need to be solved in the future. For example, Zhang et al. paid attention to the influence of features generated from different period images on extraction results, and the lack of discussion on the difference of periods, which was the focus of their later work [56]. Some researchers employed multi-spectral data, vegetation indices and phenological metrics to enrich the information available for crop mapping, which may increase computation time with little improvement in accuracy [57,58]. The method we demonstrated can solve the problems and achieve the goal of giving consideration to both periods and features and also can be applied to the problems associated with large volumes of data.

### 5.1. Winter Wheat Mapping in Heterogeneous Planting Conditions

After analysis, the heading stage was found to be the optimum period for winter wheat extraction from other land cover types in both study areas, but the selection of features showed variability. The maximum and minimum contribution of winter wheat extraction in NAC were NDVI and NDWI, while in CAC were band 6 and NDBI, respectively. Based on the Sentinel-2 image at the heading stage, the distribution of winter wheat could be achieved according to the results of feature selection. The producer's accuracy and user's accuracy of winter wheat in NAC was about 95% and 93%, respectively, while those in CAC were about 80% and 78%, respectively. The difference between the results confirmed the importance of considering the planting conditions for mapping.

### 5.2. Factors Influencing the Accuracy of Winter Wheat Mappings

The difference in mapping accuracy between the two areas may be due to the specific conditions of the two study areas. Located in the northern plain of Anhui province, NAC is a winter wheat intensive planting area with continuous spatial distribution, regular patches and large planting area. Compared with NAC, the winter wheat planting scale in CAC is relatively small and most of which are discontinuous and irregularly distributed making it difficult to identify winter wheat fields. Previous studies have shown that fragmentation has a greater impact on crop mapping [6,12]. The higher the land fragmentation level, the more serious the mixed pixels are and the lower the mapping accuracy of winter wheat is, which may partially explain the phenomenon that the extraction accuracy of winter wheat in NAC was higher than that in CAC. Moreover, crop types and planting patterns determined the complexity of winter wheat extraction. The winter crops in NAC were only barley and wheat, and the planting area of wheat was large, so barley can be neglected compared to its planting scale. The winter crops in CAC were barley, oilseed rape and wheat. The large area of oilseed rape would affect the extraction results and make the wheat mapping in CAC more complex. The mapping accuracy could be improved when there is a better method to eliminate the influence of other winter crops, such as oilseed rape.

### 5.3. Winter Wheat Mapping Using Optimum Feature Subset

In our study, the percentage error of winter wheat extraction was lower than 5% and Kappa was greater than 0.83 by using the first five features with the highest score in NAC. The percentage error of the first seven features in CAC was lower than 25%, and Kappa was greater than 0.7. The higher classification accuracy was achieved based on the optimum features of scheme D compared to those in scheme A with all indices and the features in scheme B with all spectral bands. The reason may be that the combination of optimum subsets of all types of features takes advantage of multi-source information to maximize useful information compared with a single feature. Although the highest classification accuracy was obtained using scheme C for both study areas, scheme D removed features

with low importance and only retained those contributed significantly to winter wheat extraction, so the workload was reduced and the work efficiency was improved significantly.

*5.4. Uncertainty Analysis and Future Needs*

In this study, we explored the mapping of winter wheat in heterogeneous planting conditions and got good results. However, there are still some uncertainties that need to be addressed in the future. First, this study lacks field investigation data. Second, only four winter wheat planting counties were selected as the study areas since the available images were reduced due to cloud cover and bad weather, and only one winter wheat growing season from 2017 to 2018 was selected for the study. More work needs to be conducted in the future, such as expanding the scope of the study area and choosing multiple growing seasons to verify the applicability and generalization of the conclusions in this study. Third, we only used machine learning methods to extract winter wheat in the study area. Further efforts could be implemented to evaluate the influence of different methods (such as deep learning) on the accuracy of wheat mapping.

## 6. Conclusions

An integrative analysis of the optimum period, optimum screening feature and optimum extraction scheme was explored for winter wheat mapping in Anhui province in China using high-spatial-resolution Sentinel-2 images. In both study areas, the optimum period for winter wheat extraction was the heading stage and the optimum features were NDVI, $NDVI_6$, band 11 (1614 nm), band 2 (496 nm) and band 8 (835 nm) for NAC, and band 6 (740 nm), NDVI, GNDVI, band 2 (496 nm), band 11 (1614 nm), EVI and band 4 (665 nm) for CAC. Based on the optimum feature scheme, random forest classifier generated a Kappa of about 0.85 in NAC and a Kappa of 0.75 in CAC, accompanied by a reduction of more than 60% in computational cost of image analysis. The wheat maps had high accuracies, which can support the utility of the maps for depicting the spatial distribution of winter wheat. The value of our research lies in that relatively less resources in terms of datasets and timing would be employed to obtain practical and accurate wheat planting information, so as to provide valuable references in methods and make up for the shortage of wheat study for the areas where challenged by high degree of fragmentation, complex terrain surface and changeable climate. The result provides references for agricultural and government departments to make decisions, as well as food security issues.

**Author Contributions:** D.Z. contributed to the study design and data acquisition and led the writing of the manuscript. S.F. performed the study, analyzed and interpreted the data. B.S. conceived the experiments, and responsible for the research analysis. H.Z. contributed to manuscript writing and gave advice on designing the experiments. N.J. and H.X. reviewed and revised the manuscript. Y.Y. provided technical support in remote-sensing data analysis. Y.D. contributed to literature research and data analysis.

**Funding:** This research was funded by Anhui Provincial Major Science and Technology Projects, grant number 18030701209, National Natural Science Foundation of China, grant number 41771463, 41771469, 61672032.

**Conflicts of Interest:** The authors declare no conflict of interest.

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
