# Peer review of "Winter Wheat Mapping Based on Sentinel-2 Data in Heterogeneous Planting Conditions"

_remotesensing, doi:10.3390/rs11222647_

Round 1

Reviewer 1 Report

Dear Authors,

Your manuscript entitled “Winter Wheat Mapping Based on Sentinel-2 Data in 3 Heterogeneous Planting Conditions” is a very interesting article, it has well-described methodologically steps, and reader's introduction to the whole work and analysis. In addition, the use of information from Sentinel and Planet is methodically interesting for the analysis of winter wheat.

Table 1 - the dates for CAC and NAC are the same, so it can be presented in a different way than currently presented in the table, there is no information how many scenes and what size were used in the analyzes.

In section 2.2.2 Planet Imagery - you can add a summary of the dates of acquired images for Sentinel with those obtained for Planet. If we compare information from several sensors or check them, it should be similar in time.

Line 163 - should be JM written with the full name because it is used for the first time in the text.

Line 180 – What means “optimum period imagery “?

Line 191 - why such schemes? Have any tests been done?

Table 4 - not clearly presented

Figure 4 - what does the different color on the charts mean?

Figure 7 - not clearly presented

The authors responded to the goals set in the article's introduction.

Sincerely

Reviewer

Reviewer 2 Report

The manuscript tries to present a methodology to classify wheat using Sentinel-2 data under heterogeneous planting areas. Unfortunately, the goal of the study is unclear. Part of the introduction is outdated, and the story is not clear. Methodology section need complete revision, from errors in satellite spatial resolution to tutorial like section. It is not clear what was done. Was the best date determined by the Jeffries-Matusita distance? Looks like by the results, but not said in methods section. The methodology is very hard to follow, and it is very confusing. Apparently, Planets images where used as input in a random forest algorithm (that was not described) and the classified Planets image was used as ground true to assess Sentinel-2 classification. So, looks like the authors are evaluation a remotely sensed classification based on another equally uncertain remote sensing classification. Why?

In addition, there are many unnecessary and, at a certain level, unscientific sentences throughout the manuscript e.g. ‘Currently, Planet is one of the largest satellites in orbit in the world.’ (line 140) ‘…obvious climate change.’ (line 116). Such expressions should be avoided in scientific writing.

English review is needed. Typos such as: ‘seaso’ in line 26 and other grammatical errors throughout the manuscript need correction.

Please find more specific comments below.

Abstract needs clarification, results are presented before methods. Line 26 has a result stateman and from line 27 to 34 are a summary of material and methods. Also, authors should not use abbreviations such as JM, PE, etc in the abstract section. These abbreviations were not explained and generate confusion. I also suggest to not use NAC and CAC, simply say northern Anhui counties and central Anhui counties. In general, there is no need to use abbreviations in the abstract.

Line 26-27: What sentence on line 26-27 means? What the authors means with separability and what is JM distance?

The method presented in the abstract is confusing. Please rephrase sentences from line 27 to 34. First present all material used i.e. all satellite imagery. Sentinel was used for classification and Planet was used to evaluate.

Line 67-72: This sentence is outdated. Sentinel-2 has what the authors point as expected.

Line 79-80: This states that the paper is only relevant for Anhui province. The next paragraph states a much clear importance of the study. Complex terrain, cloud cover, fragmented farming, etc. These challenges happen to be in Anhui but are valid for other regions.

Line 87-88: contradicts statement in line 79-80.

Line 94-102: This paragraph presents a mixture of objectives and methodology used. Please clearly present the main goal of the study and do not use specific objectives as a methodology section.

Table 1 has a misleading title and should be presented in the study area section.

There is no need to have separate section for Sentinel-2 and Planets dataset.

Figure 2: Abbreviations and acronyms should be explained. What is JM, MNDWI, NDBI, etc. This figure does not really help understand what was done. This table should be in the end of the methods, after the authors had explained what the intend.

Section 3.1. Authors describe SNAP, Sen2Cor, and ENVI data processing as a tutorial.

Table 2: NEVER use two abbreviations for the same thing! B = QB = Blue wavelength = MSI band 1!? Red_edge1 = r_e1 = Qr_e1 = MSI band 5. Authors should use conventional band number presented by ESA (https://sentinel.esa.int/web/sentinel/missions/sentinel-2/instrument-payload/resolution-and-swath)

Band 1 (443.9 nm) has space resolution of 60 m NOT 10 m!!

Section 3.5. Random forest is not slightly explained! Authors only say that it has been successfully used before. How the algorithm was parameterized? What number of trees and number of variable per level were used? How the algorithm was trained? Cross-validation? How?

Line 223: Why random forest information is here?!

Line 248: Planets images where used as input in a random forest algorithm and the classified Planets image was used as ground true to assess Sentinel-2 classification?! The authors are evaluation a remotely sensed classification based on another equally uncertain classification? Why? This manuscript is getting very confusing.

Figure 6 and 7: Instead of showing many maps that are visually identical it would be much better if difference maps were presented so the differences between the “true” and different features can easily be seen.

Discussion section is very repetitive with all subsections starting with summary phrases or as in Section 5.1, half of it is a repetition of introduction literature.

Line 408: relatively few resources? I do not agree. Planets image acquisition is not cheap!

Conclusion is not very conclusive since the objectives are not clear.

I recommend a full revision of the entire manuscript and if the Planets imagery were used as inputs to a machine learning classification to be used as ground true it is wrong. You can not assess a remote sensing classification method based on a equally uncertain method.

Reviewer 3 Report

The paper is focused on the mapping of winter wheat by using Sentinel 2 data in complex and heterogeneous planting conditions. The manuscript is interesting and destined to be published; however, there is a number of grammatical mistakes and style flaws to be corrected.

Firstly, both the incipit of Abstract and Discussion can be improved. In the first case, the beginning sentence is too generic and is built by using the present tense (“become”) in state of a more correct past tense. In the second case, rows spanning from 341 to 350 are crammed full of concepts already expressed.

Often, I have observed a tendency to repetition of same words in a singular sentence. Please, remove these repetitions (see line 73 in which there is a repetition of the word difference).

In Figure 1, it could be useful to show where the examined areas are located within the China.

Moreover, captions of Figures and Tables must end with full stop (see, for example, captions of Table 1 and Figure 3).

Words should not be written as bold text ( see lines 114 and 119).

Expressions such as high-resolution data, high-spatial resolution satellites should be always written by placing a dash between those two words (e.g. line 66).

Sentences should not begin with number (line 206) or with adversative conjunctions (see line 67).

In Figure 3 it could be helpful to add a bar scale to correctly interpret the size of plots.

In several cases, I have observed the use of unsuitable words (e.g. : line 42 regret in state of loss; line 116 obvious in state of affected by climate change; line 191: exam in state of examine).

Specific mistakes:

line 26: JM is cited as acronym without providing full explanation;

line 27: the cited red-edge indices are already included in the category vegetation indices;

lines 32-34: Please remove the word were to give sense to the sentence: “Five sample plots in NAC and six sample plots in CAC were derived from Planet images to evaluate the winter wheat extraction results derived from Sentinel-2 images with percentage error (PE) and confusion matrix”;

lines 48-49: for  the sentence “Wheat is the third largest food crop in term of production globally [1], providing a large number nutritional source for these suffered from nutrient deficiency” suggested alternative “Wheat is the third largest food crop in term of production globally [1], providing a large number nutritional source for these suffering from nutrient deficiency”;

lines 56-58: for  the sentence “Remote sensing technology has been widely used in the field of crop identification and mapping [5-7] due to their spatial coverage, temporal resolution, availability at near real time and low cost” ; suggested alternative “Remote sensing technology has been widely used in the field of crop identification and mapping [5-7] due to its spatial coverage, temporal resolution, availability at near real time and low cost”;

line 63: The word data is plural;

lines 91-93: “In view of the important role of Anhui province in China's wheat production and the problems in the field of remote sensing extraction of winter wheat, it is urgent to explore the winter wheat remote sensing mapping in different regions of Anhui” suggested alternative  “In view of the important role of Anhui province in China's 91 wheat production and the problems in the field of remote sensing extraction of winter wheat, it is 92 urgent to explore remote sensing-based mapping of winter wheat in different regions of Anhui”;

line 110: This sentence needs the subject: “Northern and central zones are the main producing areas of winter wheat in Anhui”;

The word Earth must be written with capital letter (see line 141);

In two cases the word “architecture” seems to be used in state of urban (see line 166);

Line 238: Kappa was calculated as….;

Line 247: Is it possible to report the wavelengths of the four spectral bands of Planet?

Line 309: A, B, C, D represent the wheat distribution based on scheme A, B, C, D. (see also a similar mistake at line 329).

Reviewer 4 Report

The present manuscript deals with the classification of winter wheat in the Anhui Province of China. 

This reviewer read the manuscript in the beginning with huge interest, but this changed when reading the objectives of the study. These objectives have already been addressed by numerous research papers in different parts of the world with just little variations. Thus, this reviewer suggests reject this manuscript.

Reviewer 5 Report

The manuscript 'Winter Wheat Mapping Based on Sentinel-2 Data in Heterogeneous Planting Conditions' has scientific relevance and is within the scope of the journal Remote Sensing. However, I still have some concerns before considering it for publication:

- better describe the methods used in the summary;
- insert the scientific name of wheat in the keywords and the first time you mention the crop in the introduction;
- it would be possible to describe the weather conditions during the experiment;
- Was any new experiment performed to validate the results obtained?
- I would like to see a boxplot for the vegetation indices studied;
- It would be possible to demonstrate the correlation between the indices used;
- The discussion is well written, but I believe it is possible to expand it and demonstrate the possible applications of the methodology proposed here.

Round 2

Reviewer 2 Report

Well done! The paper has improved considerably. It is much clear what was done and why. I commend the authors for this! However, there are some points to clarify. Optimum selection on scheme D is obscure and conclusions looks incorrect. Please see comments bellow.

Missing comma in line 20: “In this study, northern …”

Sentence suggestion in line 168: “Values of JM > 1.8 indicates good separability between two samples.” Instead of “Only when JM>1.8, indicating the separability is good between two samples.”

Figure 5 states that optimum for NAC is 5 features and 7 for CAC, so scheme D was done using the first 5 features presented in Figure 4A and the first 7 in Figure 4B. This should be stated clearly because the best result comes from scheme D!

I still think Figure 7 should be a detailed difference map. Mapped minus reference, would be easy to see which scheme presented the best results in each sample. Authors already have presented the mapped areas in Figure 6.

Line 369: Using optimum feature gave you the number of variables and which variables are the best. Present that clearly, don’t deviate saying what was slightly lower. Authors should only present the best case. “The highest accuracy was obtained using … for NAC and …. for CAC” (if different).

I am unsure if Section 5.5 should exist. It would look better as first paragraph of the Discussion, right under section 5. Discussion.

The conclusions look wrong or at least misleading! Scheme D consist in optimum features not only NDVI and band 6?! See Figure 5. Authors said that you can use NDVI for NAC and band 6 for CAC, but that would be incorrect since authors used NDVI, NDVI6, band 11, 2 and 8 for NAC and band 6, NDVI, GNDVI, band 2 and 11, EVI, and band 4 for CAC. Is that correct? Please clarify. If scheme D was only NDVI for NAC and band 6 for CAC then conclusions are correct but section 4.3 is misleading and should be corrected!

Reviewer 4 Report

please go for publication

Author Response

Thanks for your great effort on our manuscript!

Reviewer 5 Report

The authors have made the requested corrections and the manuscript can be accepted.

Author Response

(The authors gave the same response as above.)
